# Enhancement of Desulfurization Capacity with Cu-Based Macro-Porous Sorbents for Hydrogen Production by Gasification of Petroleum Cokes

**Dongjoon Kim [1,†], Dasol Bae [2,†], Yu Jin Kim [2], Seung Jong Lee [3], Jin Wook Lee [3], Yongseung Yun [3], No-Kuk Park [4,\*] and Minkyu Kim [2,\*]**

1   William G. Lowrie Chemical and Biomolecular Engineering, The Ohio State University, Columbus, OH 43210, USA; kim.7270@buckeyemail.osu.edu
2   School of Chemical Engineering, Yeungnam University, 280 Daehak-ro, Gyeongsan 38541, Korea; dasol0407@ynu.ac.kr (D.B.); rladbwls2385@ynu.ac.kr (Y.J.K.)
3   Institute for Advanced Engineering, 175-28, Goan-ro 51 Beon-gil, Baegam-Myeon, Cheoin-gu, Yongin-si 17180, Korea; sjlee@iae.re.kr (S.J.L.); jwlee@iae.re.kr (J.W.L.); ysyun@iae.re.kr (Y.Y.)
4   Institute of Clean Technology, Yeungnam University, 280 Daehak-ro, Gyeongsan 38541, Korea
\*   Correspondence: nokukpark@ynu.ac.kr (N.-K.P.); mk_kim@ynu.ac.kr (M.K.)
†   Dongjoon Kim and Dasol Bae contributed equally to this work.

**Abstract:** Macro-porous alumina was used as a support for a pellet-type Cu-based desulfurization sorbent in the gas purification process for producing blue hydrogen by the gasification of petroleum coke. The effects of the macro-porous alumina on the pellet-type sorbents in reducing the gas diffusion resistance into the pores were investigated. The results showed that the macro-porous alumina enhances the diffusion resistance, resulting in an improved sulfur capacity of CuO absorbents. Such effects were more significant on the pellet type CuO absorbents than the powder type. In addition, CO production was observed experimentally during the desulfurization reaction of carbonyl sulfide (COS) at low temperatures (~473 K). Density functional theory calculations were also performed to understand the kinetics of desulfurization and CO production. The simulation results predicted that the kinetics of desulfurization is strongly affected by the local surface environment. The CO generated from C–O bond breaking from COS had a lower adsorption energy than the $CO_2$ formation. These results suggest that the Cu-based desulfurization sorbent has potential catalytic activity for producing CO from COS dissociation.

**Keywords:** COS removal; desulfurization; Cu-based sorbent; macro-porous alumina; COS dissociation

## 1. Introduction

Global warming is putting a brake on the continuous use of fossil fuels used in industry. The global energy market is demanding a transition to renewable energy that does not emit carbon dioxide [1]. Nevertheless, the infrastructure for renewable energy is insufficient. The fact that most of the energy used in the industry depends on fossil fuels means completely converting to renewable energy in a short time will be a daunting task. Therefore, it is necessary to develop a technology that can reduce or block carbon dioxide emissions from the fossil fuel-based energy conversion technology currently used. From this point of view, the production of blue hydrogen is an important energy issue [2]. Blue hydrogen refers to hydrogen produced in a process that does not emit carbon dioxide to the atmosphere via the capture and storage of carbon dioxide generated by hydrogen production from conventional fossil fuels.

Hydrogen production technology by petroleum coke gasification is a promising technology for the mass production of blue hydrogen [3–5]. Hydrogen production technology by petroleum coke gasification can be divided into gasification technology, gas purification technology, gas conversion, and separation. Syngas produced by the gasification of

petroleum coke contains various impurities, including sulfur compounds, chlorides, nitrogen compounds, such as ammonia, and heavy metals. Among these impurities, gaseous sulfur compounds, such as $H_2S$ and COS are emitted. COS can be converted to hydrogen sulfide in the catalytic hydrolysis process. It is removed selectively using wet amine process and iron chelate process. High purity hydrogen can be produced by processes where purified syngas passes through the water gas shift (WGS) reaction and the pressure swing adsorption (PSA) separation process. A large amount of carbon monoxide in synthesis gas is converted to hydrogen by the water gas shift reaction. The hydrogen and carbon dioxide generated are separated in the PSA process to produce high-purity hydrogen. Importantly, when trace amounts of sulfur compounds are present in the synthesis gas, they deactivate the catalyst (generally $CuO/ZnO/Al_2O_3$ catalysts) for the water gas shift reaction due to the sulfidation of active sites. Because of the deactivation issue, a desulfurization process is a critical and essential step to produce hydrogen gas from petroleum coke.

Various types of sorbents including activated carbon, metal oxides, zeolite, composite oxides, metal-organic frameworks, etc. have been extensively studied [6–8]. Different types of absorbent have different strength and weakness. For example, metal oxides have shown high reactivity and high sulfur capacity; however, metal oxides need to be regenerated at high temperature and it triggers aggregations deactivating the absorbents [9–12]. Compared to the other metal oxides absorbents, Cu-based sorbents have been proposed to have higher desulfurization performance than ZnO, $Fe_2O_3$, $MnO_2$, and CaO at a low temperature (<400 °C) owing to their thermodynamic equilibrium [8,13]. Recently, studies on desulfurization sorbents with high sulfur capacity for the removal of low concentration sulfur compounds using mesoporous materials and MOFs as supports as Cu-based desulfurization sorbents have been reported. Absorbents with a high surface area are required to remove low concentrations of sulfur compounds effectively. Hence, mesoporous materials with high surface areas have been evaluated as support materials [12–21]. Although Cu-based sorbents with a support improve the surface area and show good desulfurization performance, they generally have high gas diffusion resistance reducing the desulfurization capacity. Gas-phase molecules containing sulfur adsorb on the outer surface, but unreacted active sites can remain inside the absorbent due to the high gas diffusion resistance.

This study developed a macro-porous alumina support to enhance gas diffusion to resolve the above mentioned issue of gas diffusion resistance. CuO absorbents were supported on commercial $\gamma$ alumina and macro-porous alumina prepared by an impregnation method. In particular, macro-porous alumina has been synthesized with nano-sized colloidal particles (template) by the suspension polymerization of a polymer, such as poly-methyl-meta-acrylate (PMMA) and polystyrene (PS). The colloidal particles used as a template are mixed with an alumina precursor solution. The moisture is then removed by vacuum evaporation. The particles can be prepared on the macro-porous alumina by oxidatively decomposing the polymer by heat treatment. In previous studies, macro-porous materials were synthesized using polymer beads as templates for catalyst materials, such as silica, titania, and alumina [22–25]. With these $\gamma$-alumina and macro-porous alumina supports, pellet and powder types of Cu-based absorbents were prepared. The resulting absorbents were tested for the removal of COS, and the sulfur capacities of the absorbents were evaluated.

The experimental results showed that macro-porous alumina-based CuO absorbents provide better sulfur capacity than the $\gamma$ alumina-based CuO absorbents (~1.3 times higher sulfur capacity). This suggests that the enhancement of gas diffusion resistance by structural modification improves the sulfur capacity. In addition to the enhancement, CuO produces CO from COS at low temperatures (~473 K). A computational study was also performed using density functional theory calculations (DFT) to understand the surface kinetics of COS removal and CO production on the CuO surface. The computational results show that sulfur removal from COS is strongly affected by surface environments, such as oxygen vacancies and surface sulfur. Moreover, the stability of CO generated from the C–O bond cleavage of COS was lower than the $CO_2$ formation energy. This suggests that CO

prefers to desorb from the surface rather than form $CO_2$. Such kinetics would occur more frequently at low temperatures because the reaction rates are distinguished more clearly at low temperatures.

## 2. Experimental Details

### 2.1. Preparation of PMMA Colloidal Solution by Suspension Polymerization

In this study, PMMA beads dispersed in a PMMA colloidal solution were used as a template for preparing macro-porous alumina. A PMMA colloidal solution was prepared from methyl-meta-acrylate (MMA) by suspension polymerization. Distilled water (900 mL) was placed in a round-bottomed flask with a reflux condenser installed at the top. PMMA was synthesized in the polymerization apparatus purged with nitrogen gas. After injecting 150 mL of MMA solution through the stem into the flask, heating to 70 °C, and stirring at approximately 800 rpm, 0.6 g of 2,2'-azobis dihydrochloride was injected as an initiator for polymerization. Polymerization proceeded for approximately 8 h. Spherical particles, approximately 200 nm in size, were formed in the polymerized PMMA colloidal solution, which was confirmed by SEM (Figure 1).

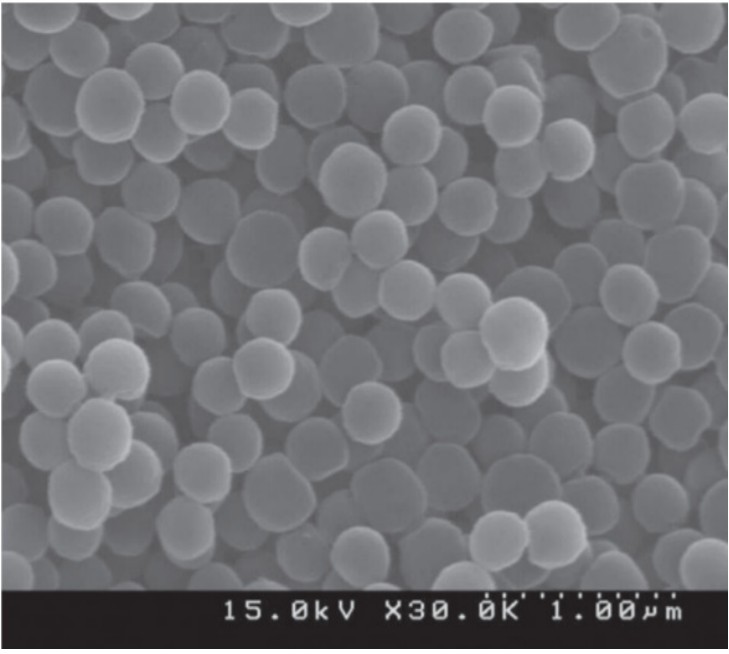

**Figure 1.** SEM image of nano-sized PMMA spherical beads synthesized by suspension polymerization with MMA.

### 2.2. Preparation of Macro-Porous Alumina by Template Method

Macro-porous alumina was prepared by mixing aluminum nitrate and a PMMA colloidal solution and heat-treating the solid obtained by removing moisture in a rotary vacuum evaporator at 600 °C. At this time, the concentration of alumina precursor was approximately 1.0 M, and the volumetric mixing ratio of the precursor and the PMMA colloidal solution was approximately 1:4. The concentration of the precursor and the mixing ratio of the PMMA colloidal solution for preparing the macro-porous alumina were prepared using the method optimized in a previous study [22–25]. As shown in Figure 2, when the alumina precursor solution and the PMMA colloidal solution were mixed, and moisture was removed by vacuum evaporation, the aluminum precursor remained as a solid, in which the PMMA beads were stacked and filled the interstices of the PMMA beads. Macro pores were formed in the structure of the alumina solid when the spherical PMMA was decomposed thermally and oxidatively by heat treatment.

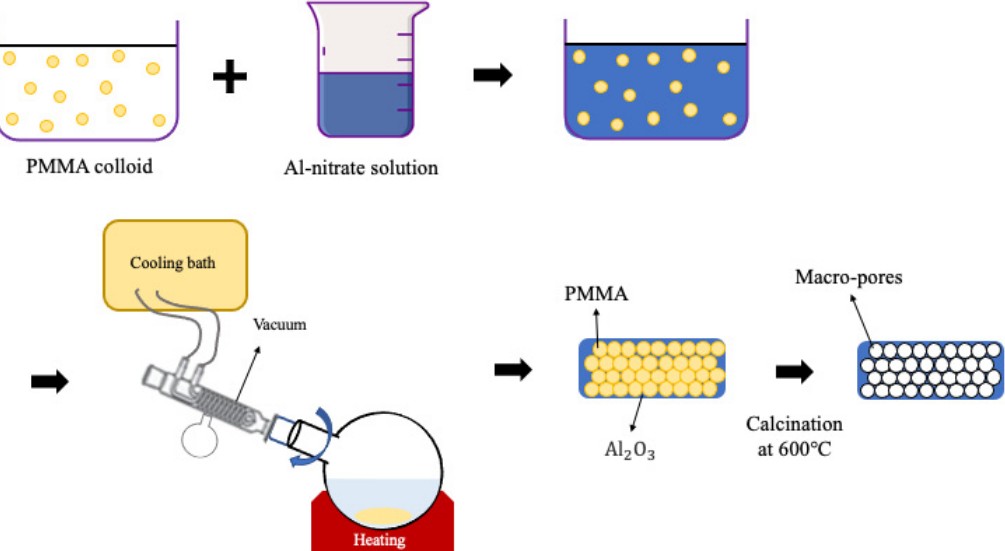

**Figure 2.** Procedure for the preparation of macroporous alumina by template method.

### 2.3. Preparation of CuO/Alumina Absorbent by Impregnation Method

CuO was deposited on the macro-porous alumina and commercial alumina powder by impregnating copper nitrate using a rotary vacuum-evaporator. The amount of CuO supported on $\gamma$-alumina and macro-porous alumina was varied (5, 10, and 15 wt %). After supporting the Cu precursor, a $CuO/Al_2O_3$ absorbent was prepared by heat treatment at approximately 600 °C. The heat-treated Cu-based absorbent was pulverized into a fine powder to form pellets, and approximately 5 wt % of methylcellulose was added as an organic binder for molding, and approximately 5 wt % of calcium silicates as an inorganic binder. After kneading by adding water, it was shaped by an extrusion method, and a pellet-type absorbent was prepared, as shown in Figure 3.

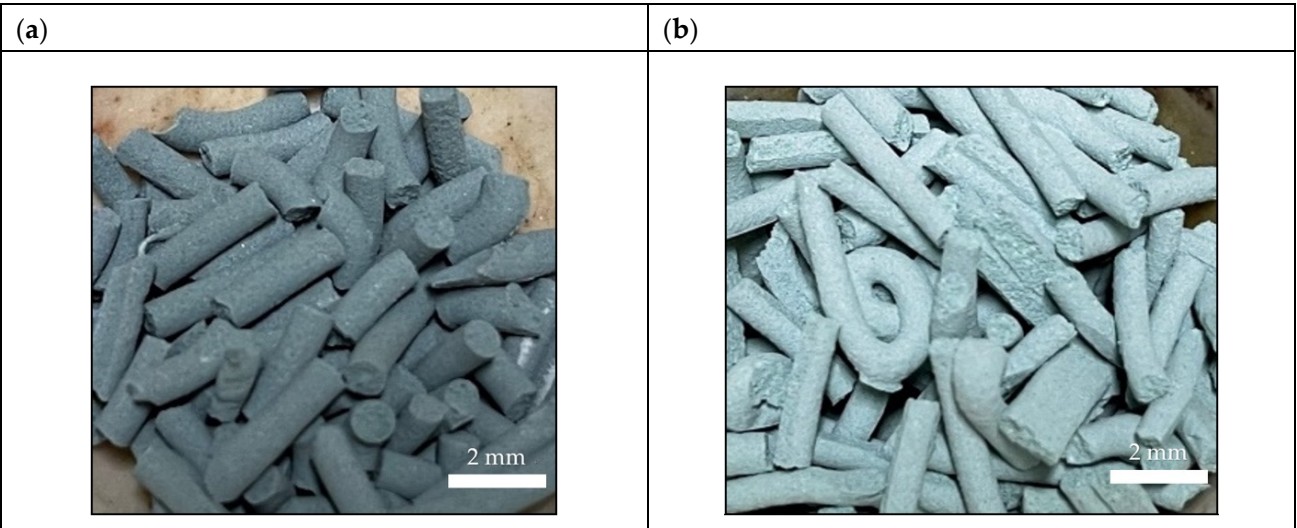

**Figure 3.** Images of the pellet-type Cu-based desulfurization sorbents shaped by the extrusion method, (**a**) $CuO/\gamma$-$Al_2O_3$, (**b**) $CuO/$macro-porous $Al_2O_3$.

### 2.4. Desulfurization Tests of Absorbents

The performance tests of the desulfurization absorbent were carried out in a fixed bed reactor, as shown in Figure 4. The experimental apparatus consisted of a gas supply device, a 1/2-inch quartz tube reactor installed vertically in an electric furnace and a gas analyzer. The content of COS flowing into the reactor was diluted with nitrogen gas, and

the flow rate of all gases was adjusted using a mass flow controller (MFC). The Cu-based absorbents supported on γ-alumina and the macro-porous alumina were filled with the powder or pellet sample in the center of the reactor, and the change in COS content was measured by gas chromatography connected to the reactor outlet after supplying the COS gas. The reaction temperature was fixed at 200 °C, and a thermocouple was installed directly in the absorbent packed bed inside the reactor to measure the temperature of the absorbent layer. For gas chromatography (GC, Shimadzu GC-8A), the multi-packed column, which was connected to Parapak T (2 ft) and Hayasep Q (8 ft) as GC column material capable of separating nitrogen, COS, and CO, was used. The column temperature was set to 140 °C, and the injector and thermo-conductivity detector (TCD) temperatures were set to 150 °C. The concentration of COS supplied into the reactor was approximately 5000 ppmv and approximately 3000 ppmv in the case of powder absorbent and pellet absorbent, respectively. In the experimental process of the pellet-shaped absorbent, the error due to the channeling phenomenon caused by the difference in the packing density of the absorbent was minimized by flowing a low COS concentration. Moreover, with the powder absorbent sample, approximately 1.0 g was packed in the reactor, and in the case of the pellet-type absorbent, and 2.0 g was packed to conduct the reaction experiment. The sulfur capacity of the sorbents was calculated based on the time in which COS was detected at the reactor outlet. The desulfurization experiment was carried out for approximately 2 h after COS detection. The sulfur capacity was calculated as in Equation (1).

$$\text{Sulfur capacity, mgS/g} - \text{sorbent} = (F_{COS} \times M.W_{Sulfur})/W_{sorbent} \times BT \times 1000 \quad (1)$$

$F_{COS}$: mole flow rate of COS, mol/min
$M.W_{Sulfur}$: gram mole mass of sulfur, 32 g/mol
$W_{sorbent}$: weight of sorbents packed in the reactor, g
BT: breakthrough time, min

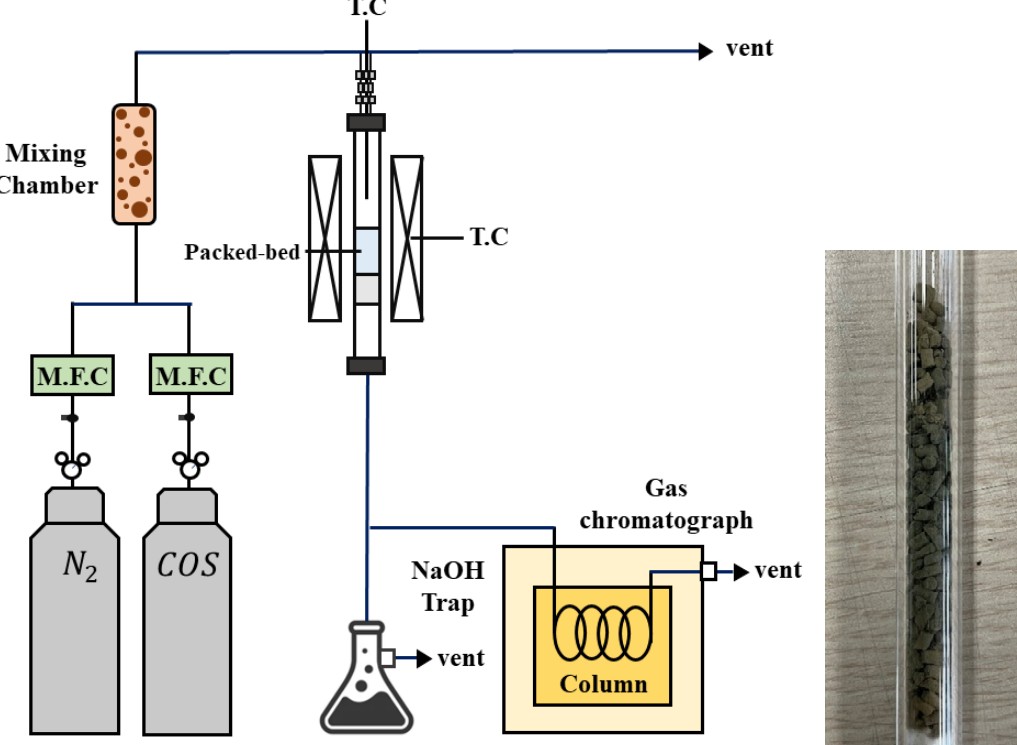

**Figure 4.** Schematic diagram of the experimental setup for the desulfurization tests.

## 2.5. Characterization

In order to investigate the physicochemical properties of the Cu-based absorbent, the surface area and pore properties by the nitrogen adsorption method (Quantachrome Instruments, AUTOSORB-1) were analyzed using Brunauer–Emmett–Teller (BET) and Barrett–Joyner–Halenda (BJH) methods. In addition, X-ray diffraction (XRD, Rigaku, D/MAX-2500, Tokyo, Japan) was used to confirm the crystal structure of the prepared absorbent. Finally, the surface shape and macropores of the absorbent were confirmed by scanning electron microscopy (SEM, Hitachi, S-4100, Tokyo, Japan).

## 3. Computation Details

All plane wave DFT calculations were performed using the projector augmented wave pseudopotentials [26] provided in the Vienna ab initio simulation package (VASP) [27]. The Perdew–Burke–Ernzerhof exchange-correlation with Hubbard U corrections [28,29] were used with a plane wave expansion cutoff of 600 and 400 eV for the bulk and surface, respectively. We did not employ dispersion corrections because we found that the energetics of small molecules such CO and S were well predicted by DFT without D3 [30]. We believe that COS can be affected by the presence of dispersion corrections; however, the proposed COS absorption mechanism would not be changed with the dispersion interactions because the dispersion effects are canceled out when evaluating activation energy barriers [31,32]. Various U values were tested to find a U value providing the proper lattice parameters of CuO. The tests showed that the U value of 8.5 eV provides proper lattice parameters of CuO, which agreed well with the experimental results. Based on the results, U of 8.5 eV was used for the computation. The C2/c space group was used with monoclinic crystal structure [33] to relax the bulk lattice constants until the forces reached at least 0.01 eV/Å with the Monkhorst–Pack KPOINTS mesh, $8 \times 8 \times 8$. The PBE+U relaxed bulk lattice constants of CuO (a = 4.754 Å, b = 3.424 Å, c = 5.178 Å, $\beta$ = 99.25°) were used to fix the lateral dimensions of the CuO(111) slab. These DFT-predicted CuO bulk lattice parameters agreed well with the experimental bulk lattice parameters (a = 4.682 Å, b = 3.424 Å, c = 5.127 Å, $\beta$ = 99.42°) [34]. Seventeen valence electrons were introduced including the 3p states in Cu for bulk and surface calculations because PBE failed to predict the experimental bulk lattice parameter without 3p electrons. This study focused on the CuO(111) surface, which is one of the three stable surfaces (($111$), ($\bar{1}11$), and ($011$)) with a CuO crystal morphology [35]. The CuO(111) surface had a $3 \times 3$ surface unit cell size with ~9 Å slab thickness. The surface consists of five layers with two fixed bottom layers. For the slab calculations, the interatomic force, 0.05 eV/Å, and $2 \times 2 \times 1$ KPOINTS mesh were used. A vacuum spacing of ~20 Å, which is sufficient to reduce the periodic interaction in the surface normal direction, was used. The pristine bare CuO(111) surface has two active sites of coordinatively unsaturated Cu ($Cu_{cus}$) and saturated Cu sites ($Cu_{sat}$) on the CuO(111) surface. The surface oxygen and subsurface oxygen were also labeled as $O_{surf}$ and $O_{slab}$, respectively. These sites are illustrated in Figure 5a. The effects of surface sulfur and surface oxygen vacancies on COS oxidation reaction toward $CO_2$ were also examined by introducing an $O_{cus}$ vacancy and a S surface to the CuO(111) surface. Figure 5b–d shows the described O vacancy and surface S atom-employed CuO(111) surfaces. All relaxed configurations of the adsorbed COS, CO, and $CO_2$ on the CuO(111) surface with S and oxygen vacancies are provided in this manuscript. Unless otherwise noted, the DFT calculations were performed for a single molecule adsorbed within the surface models described above.

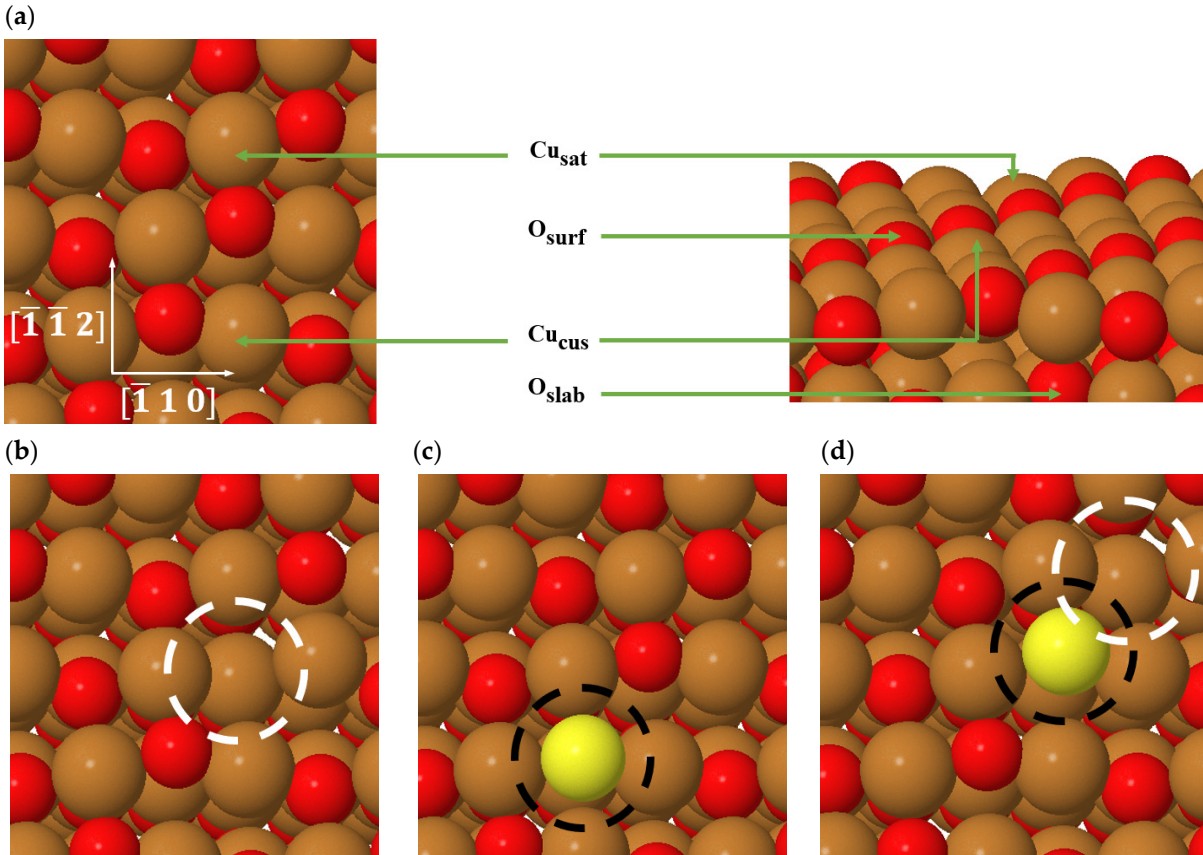

**Figure 5.** Top and side views of the (**a**) CuO(111) surface. The cus and sat on CuO(111) represent a coordinatively unsaturated site and saturated site, respectively. The vertical and horizontal arrows on the CuO(111) surface in (a) represent the [$\overline{1}\overline{1}2$] and [$\overline{1}10$] crystallographic directions. (**b**–**d**) The top views of O vacancy, surface S atom, and both O vacancy and surface S atom employed on CuO(111) surfaces, respectively. Each O vacancy (white) and surface S atom (black) are indicated by a dashed circle.

In this present study, we define the adsorption energy between a molecule and surface using Equation (2):

$$E_{ads} = E_{slab} + E_{iso} - E_{slab+ads} \tag{2}$$

where $E_{slab}$, $E_{iso}$, and $E_{slab+ads}$ are the energy of a bare surface, isolated molecule, and a molecule adsorbed on the bare surface, respectively. In this adsorption energy, a larger positive energy indicates higher stability of the adsorbed molecule under consideration. The barriers for the COS oxidation reaction toward CO$_2$ on CuO(111) were examined using the climbing nudged elastic band (cNEB) [36] method and confirmed that the resulting transition states had one imaginary vibrational frequency. All energies reported in this paper were corrected for zero-point vibrational energy.

## 4. Experimental Results

### 4.1. Crystal Structure Analysis of Cu-Based Absorbents on Γ-Alumina and Macro-Porous Alumina

The crystal structures of the prepared CuO/γ-Al$_2$O$_3$ and CuO/macro-porous Al$_2$O$_3$ absorbents were analyzed by XRD. The CuO phases were present on the γ-alumina and macro-porous alumina surfaces (see Figure 6). The XRD patterns of the CuO crystal showed peaks at 35°, 39.4°, 49°, 54°, 58°, and 61.8° 2θ. The XRD peak intensity of CuO supported on the γ-alumina surface was stronger than that on the macro-porous alumina surface. The peak patterns of CuO supported on the surface of the macroporous alumina did not appear clearly. This suggests that CuO is distributed widely on the alumina surface. As the content of CuO supported on alumina was decreased, the intensity of the crystalline

peak became lower because a low CuO content was well dispersed over a large surface of the alumina supports, which resulted in little CuO crystal growth on the supports. In contrast to CuO, the XRD pattern of alumina revealed an amorphous structure, and the peak intensity of γ-alumina was stronger than that of macro-porous $Al_2O_3$.

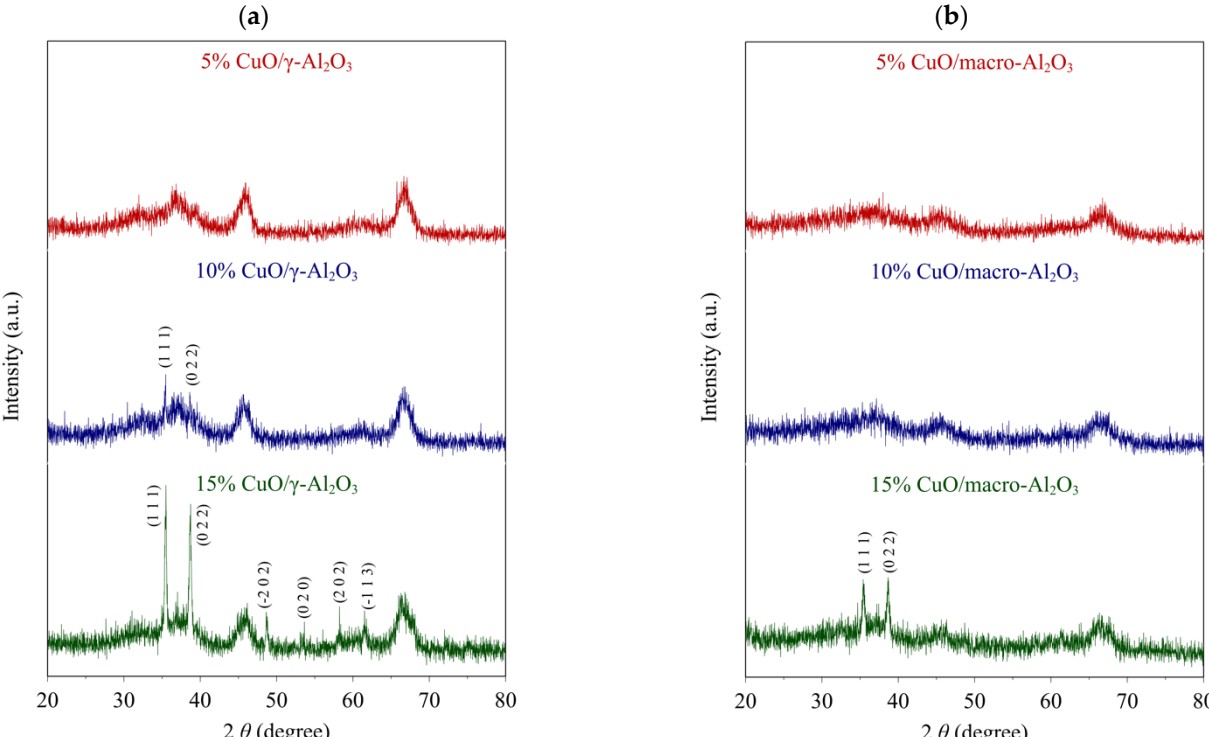

**Figure 6.** XRD patterns of Cu-based desulfurization sorbents: (**a**) 5, 10, 15 wt % CuO/γ-$Al_2O_3$; (**b**) 5, 10, 15 wt % CuO/macro-porous $Al_2O_3$.

### 4.2. Characterization of CuO on γ-Alumina and Macro-Porous Alumina

BET analysis with $N_2$ adsorption was performed to characterize the CuO on the γ-alumina and macro-porous alumina. The experiments confirmed that meso-sized pores had been formed in the γ-alumina and macro-porous alumina. As shown in Figure 7, although the amounts of CuO were different for all six samples, the hysteresis curves were observed in the adsorption and desorption isotherms. The nitrogen adsorption over the γ-alumina support was higher than that of the macro-porous alumina, and the nitrogen adsorption amount decreased with increasing amount of CuO supported. In particular, the nitrogen adsorption amount of the sample with a CuO content of 15 wt % was lower than that loaded with 5 and 10 wt % CuO. These adsorption and desorption isotherms correspond to type IV isotherms proposed by IUPAC. These can be considered as having micropores and typical cylindrical or ink bottle-shaped pores showing hysteresis curves due to capillary condensation. In the nitrogen adsorption process, the amount of nitrogen adsorbed by multi-molecular layer adsorption increased with increasing relative pressure $(P/P_0)$. On the other hand, in the process of decreasing the relative pressure $(P/P_0)$, the condensed nitrogen was desorbed gradually until the nitrogen condensed in the pore inlet had been completely desorbed. Therefore, the observed hysteresis stems from the condensation of nitrogen adsorbed in the capillary type of pores. The area of the hysteresis curve shown in the adsorption and desorption isotherms of γ-alumina was wider than that of macro-porous alumina, which was attributed to the larger pore volume of γ-alumina. The pore size of the macro-porous alumina was smaller than that of γ-alumina. Table 1 lists the pore volumes and surface area calculated using the BJH equation. The results show that the pore volume and size of γ-alumina were higher than that of macro-porous

alumina. On the other hand, the surface area and pore volume decreased with increasing amount of CuO on the support, but the pore size did not change significantly.

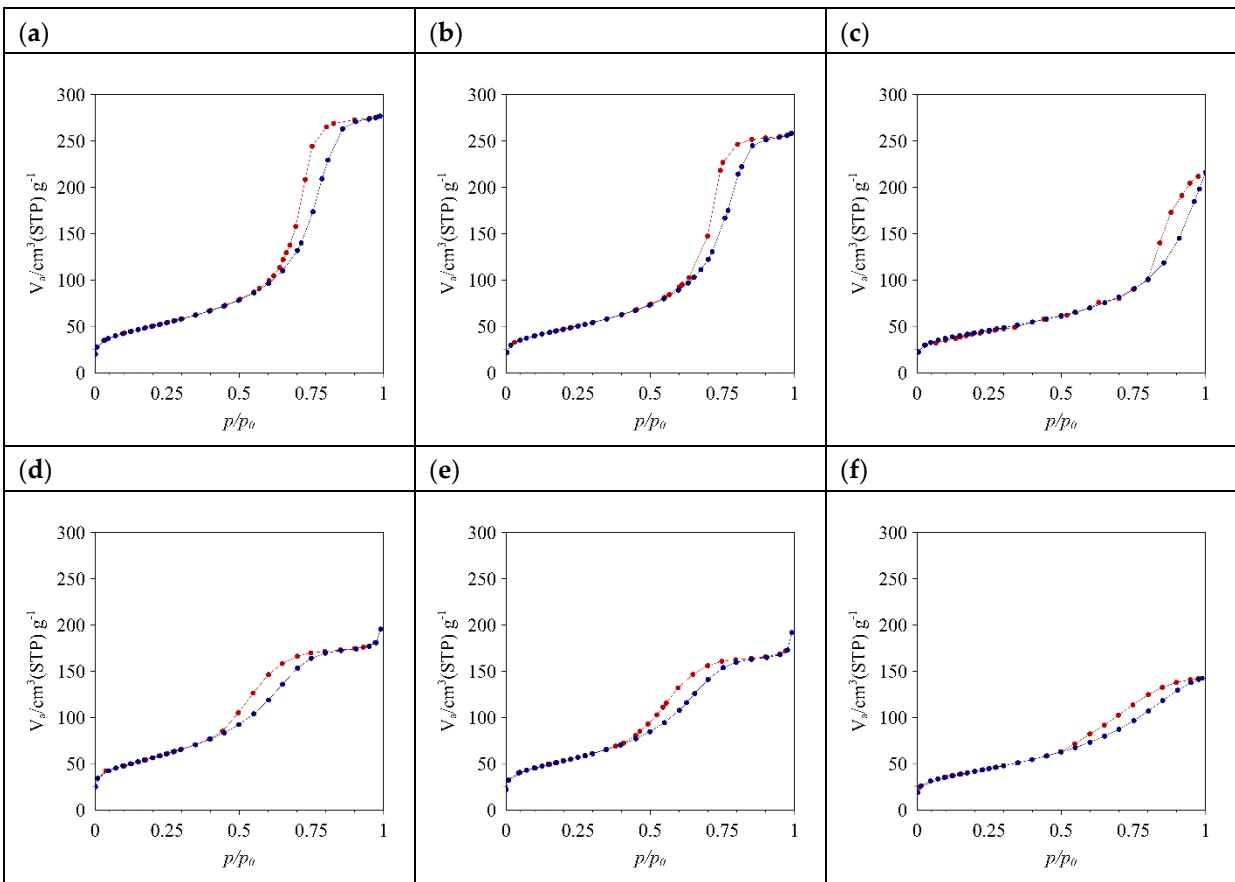

**Figure 7.** Adsorption (blue) and desorption (red) isotherm of Cu-based desulfurization sorbents by the $N_2$-adsorption method: (**a**) 5 wt % CuO/$\gamma$-Al$_2$O$_3$; (**b**) 10 wt % CuO/$\gamma$-Al$_2$O$_3$; (**c**) 15 wt % CuO/$\gamma$-Al$_2$O$_3$; (**d**) 5 wt % CuO/macro-porous Al$_2$O$_3$; (**e**) 10 wt % CuO/macro-porous Al$_2$O$_3$; and (**f**) 15 wt % CuO/macro-porous Al$_2$O$_3$.

**Table 1.** Surface and pore properties of the Cu-based desulfurization sorbents by the $N_2$-adsorption method.

| Samples | Amounts of CuO, wt % | Surface Area, m$^2$/g | Total Pore Volume, cm$^3$/g | Pore Diameter, nm |
|---|---|---|---|---|
| | 5.0 | 180.5 | 0.4280 | 9.5 |
| CuO/$\gamma$-Al$_2$O$_3$ | 10.0 | 168.2 | 0.3992 | 9.5 |
| | 15.0 | 151.1 | 0.3213 | 8.5 |
| | 5.0 | 202.8 | 0.3020 | 6.0 |
| CuO/macro-porous-Al$_2$O$_3$ | 10.0 | 188.3 | 0.2959 | 6.3 |
| | 15.0 | 148.5 | 0.2202 | 5.9 |

### 4.3. Surface Structure Analysis

Figure 8 shows the SEM results for the surface structure of the absorbent containing CuO on the surface of $\gamma$-alumina and macro-porous alumina. Macropores were dispersed uniformly over the prepared macro-porous alumina surface. As expected, such large pore sizes were not observed over the $\gamma$-alumina, and the macropores are the morphology in which PMMA used as a template was decomposed and removed. On both supports, the CuO absorbent was seen at high CuO loadings (>10 wt %). Morphologies that can be regarded as crystal growth are observed over the surface of the samples loaded with

15 wt % CuO. These crystal structures of CuO were confirmed by XRD (see Figure 6). When CuO was supported up to approximately 10 wt %, CuO was present in the dispersed state on the surface of the supports, but in the case of the 15 wt % loading, CuO crystals with a needle shape formed during the thermal treatment. The surface area of CuO corresponding to the active site of the absorbent decreased due to the growth of CuO crystals, which increased the performance as a sulfur absorbent. Overall, ~10 wt % of CuO on the supports would provide the optimal sulfur absorption capacity because of the uniform dispersion of CuO on the supports.

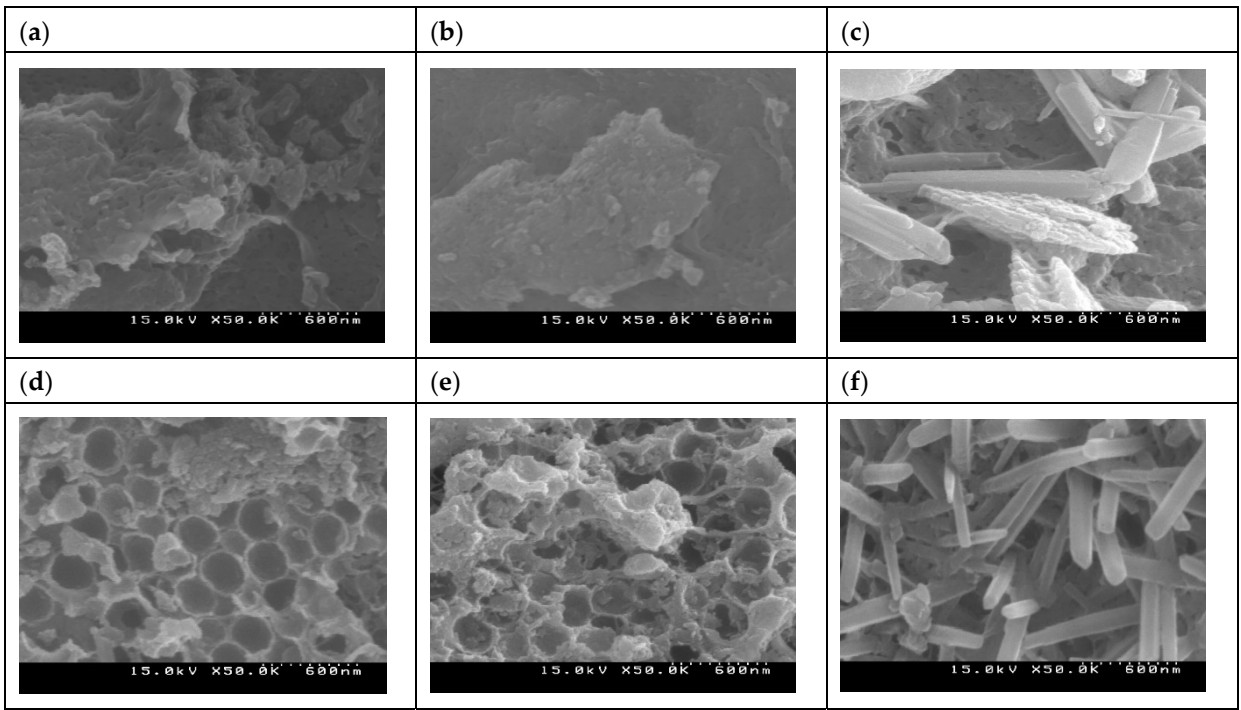

**Figure 8.** SEM images of Cu-based desulfurization sorbents: (**a**) 5 wt % $CuO/r-Al_2O_3$; (**b**) 10 wt % $CuO/r-Al_2O_3$; (**c**) 15 wt % $CuO/r-Al_2O_3$; (**d**) 5 wt % CuO/macro-porous $Al_2O_3$; (**e**) 10 wt % CuO/macro-porous $Al_2O_3$; (**f**) 15 wt % CuO/macro-porous $Al_2O_3$.

*4.4. Desulfurization Tests*

The sulfur capacity of the Cu-based COS absorbent was determined by measuring the COS concentration at the outlet of the packed bed reactor with respect to the time evolution. As shown in Figure 9a, when approximately 1.0 g of a powder type absorbent with approximately 15 wt % CuO was charged, and approximately 5000 ppmv COS was supplied at a flow rate of about 150 mL/min, the COS breakthrough time with $CuO/\gamma-Al_2O_3$ and CuO/macro-porous $Al_2O_3$ were 35 min and 40 min, respectively. The sulfur capacity of the CuO/$\gamma$-alumina and CuO/macro-porous alumina absorbed to the breakthrough time, in which COS was detected in the reactor outlet, was 37.5 mgS/g-sorbents and 42.9 mgS/g-sorbents, respectively. The sulfur capacity of the CuO/macro-porous alumina was higher than on $CuO/\gamma-Al_2O_3$, and a difference in the slope of the breakthrough curve was observed. The slope of the CuO/$\gamma$-alumina increased more rapidly than the slope of the CuO/macroporous alumina. A short breakthrough time and a sharp rise in the slope of the breakthrough curve indicated that the sulfur absorption performance of the absorbent (CuO/$\gamma$-alumina) decreased rapidly. The low performance of CuO/$\gamma$-alumina can be attributed to the high gas diffusion resistance to the pores in the relatively high-density $\gamma$-alumina.

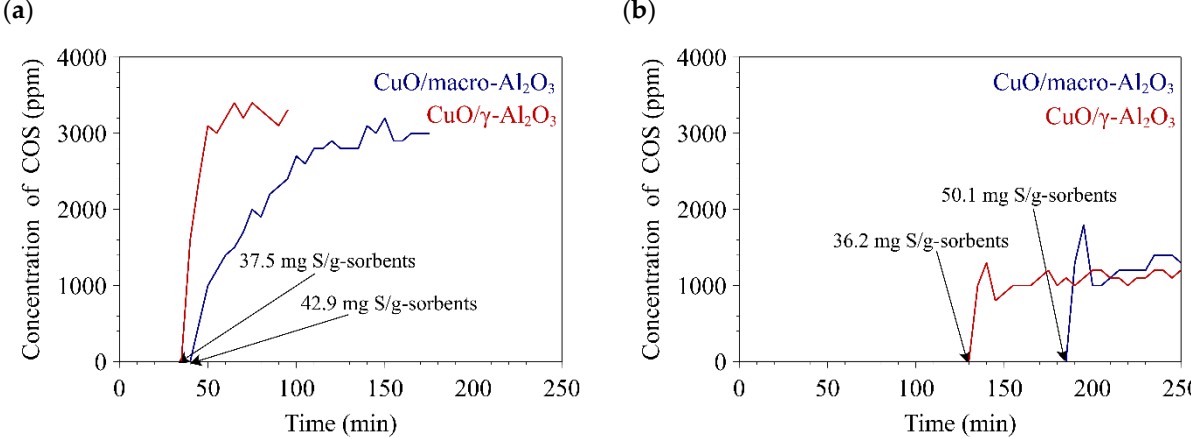

**Figure 9.** COS breakthrough curves by the desulfurization tests over (**a**) powder type and (**b**) pellet type of 15 wt % Cu-based desulfurization sorbents; CuO/γ-Al$_2$O$_3$ and CuO/macro-porous Al$_2$O$_3$.

Figure 9b presents the breakthrough curve according to the sulfur absorption of the pellet-type absorbent. These COS absorption tests of the samples (pellet type) of two absorbents with a CuO loading (approximately 15 wt %) were carried out under the experimental condition, in which approximately 3000 ppmv COS was supplied to the reactor at 130 mL/min. The sulfur capacity of γ-alumina at the breakthrough time was approximately 36.2 mgS/g-sorbents, and the sulfur capacity of macro-porous alumina showed a significant difference of approximately 50.1 mgS/g-sorbents. The gas diffusion resistance into the sorbent was higher for the pellet-shaped absorbent than the powder-type sorbent because the pellet-shaped absorbent has a large particle size. Table 2 lists the sulfur adsorption capacity of the absorbents with γ-alumina support and macro-porous alumina support with varying amounts of CuO. The sulfur adsorption capacity increased with increasing CuO content, but the sulfur capacity did not increase linearly with increasing CuO content. XRD and SEM showed that the crystal growth of CuO influenced these results because CuO crystal growth leads to a decrease in the surface area of CuO that can participate in the reaction.

**Table 2.** Sulfur capacity of the Cu-based desulfurization sorbents according to the CuO content.

| Samples | Amounts of CuO, wt % | Sulfur Capacity, mgS/g-Sorbent |
|---|---|---|
| CuO/γ-Al$_2$O$_3$ | 5.0 | 19.5 |
| | 10.0 | 43.2 |
| | 15.0 | 36.2 |
| CuO/macro-porous-Al$_2$O$_3$ | 5.0 | 26.5 |
| | 10.0 | 48.8 |
| | 15.0 | 50.1 |

CO production was observed during the sulfur absorption test of the Cu-based absorbent, as shown in Figure 10. The concentration of CO in the reactor outlet increased to approximately 1900 ppmv after the start of the COS absorption test and was maintained at approximately 1200 ppmv as the reaction proceeded. After COS was detected at the reactor outlet, the CO concentration decreased with increasing COS concentration in the reactor outlet. This result differed from the known conversion of oxygen in the metal oxide and sulfur in COS to metal sulfide and carbon dioxide. This suggests that the COS dissociation reaction occurs along with the sulfur absorption by the gas-solid reaction of the metal oxide and COS. In addition, the concentration of COS was maintained without increasing rapidly to the initial concentration after the breakthrough time of COS. This was attributed

to the dissociation reaction of COS that keeps occurring after the breakthrough curve. The generation of CO by the dissociation reaction of COS means that the Cu-based absorbent is not simply an absorbent but potentially has catalytic activity to produce CO from COS(g). DFT calculations of the desulfurization kinetics for COS on a CuO surface were performed to unveil the origin of CO production and the desulfurization mechanism.

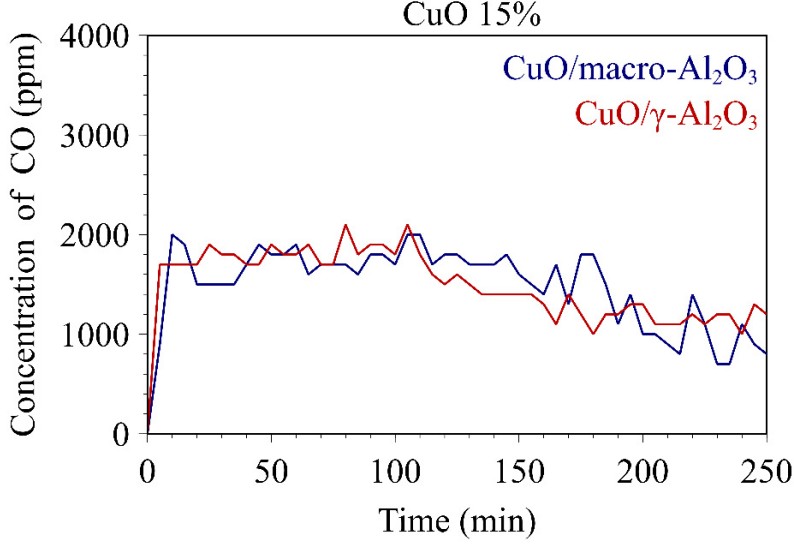

**Figure 10.** CO concentration vs. time by desulfurization tests over pellet type of 15 wt % CuO-based desulfurization sorbents: CuO/$\gamma$-Al$_2$O$_3$ and CuO/macro-porous Al$_2$O$_3$.

## 5. Computational Results

### 5.1. Stability of Desulfurization Intermediates

The simulations first evaluated the stability of CO$_2$*, COS*, and CO* on pristine CuO(111) where * represents a molecule adsorbed on the surface. The Cu$_{sat}$ and Cu$_{cus}$ top sites on CuO(111) were compared to investigate the favorable adsorption site for each adsorbate. DFT predicts that both COS* and CO$_2$* do not adsorb on the Cu$_{sat}$ site but are instead pushed and converged toward the Cu$_{cus}$ site. The CO* on Cu$_{cus}$ was predicted to be ~24 kJ/mol more stable than CO* on Cu$_{sat}$. Therefore, all the proposed results are based on the molecules adsorbing on Cu$_{cus}$ site. Figure 11a,e,i shows the relaxed adsorbates of CO$_2$*, COS*, and CO* on the pristine CuO(111). Each adsorption energy could be found immediately below at each relaxed image in Figure 11. The DFT results show that the desulfurization intermediates of CO$_2$*, COS*, and CO* bind weakly to the CuO(111) surface; the predicted adsorption energies are <35 kJ/mol. In particular, the low stability of COS* would prefer to desorb from the surface and rarely lead to activating the initial desulfurization step for CO–S bond dissociation if the reaction has a relatively large energy barrier. In comparison with the experimental data, the predicted weak binding of COS does not correspond to the high desulfurization performance of CuO. This study focused on other surface environments, including a defect surface and a sulfurized surface potentially generated during the desulfurization reactions to resolve the inconsistency. For the defect surface, the defect surface was modeled by introducing a surface oxygen vacancy and evaluating the stabilities of the adsorbed CO$_2$*, COS*, and CO* on the CuO(111) surface. Figure 11b,f,j presents the relaxed adsorbate images with oxygen vacancies; the vacancy site is illustrated by a white dashed circle. The CO$_2$* (9.0 kJ/mol) still binds weakly to an oxygen vacancy on CuO(111). In contrast to CO$_2$*, an adjacent oxygen vacancy stabilizes the adsorbed CO*(99.7 kJ/mol) and COS*(29.3 kJ/mol). The oxygen vacancy defect provides high stability with CO*, and the results suggest that the presence of oxygen vacancies on the surface provides active sites for the CO molecule. The adjacent oxygen vacancy affects the electronic structures of the adjacent Cu$_{cus}$ site significantly. It would trigger stronger back donation from Cu$_{cus}$ to the adsorbed CO* molecule, thereby providing the

high stability of adsorbed CO* [37]. Such effects of oxygen vacancies have been reported in other reactions on CuO surfaces and other transition metal oxides [37–39].

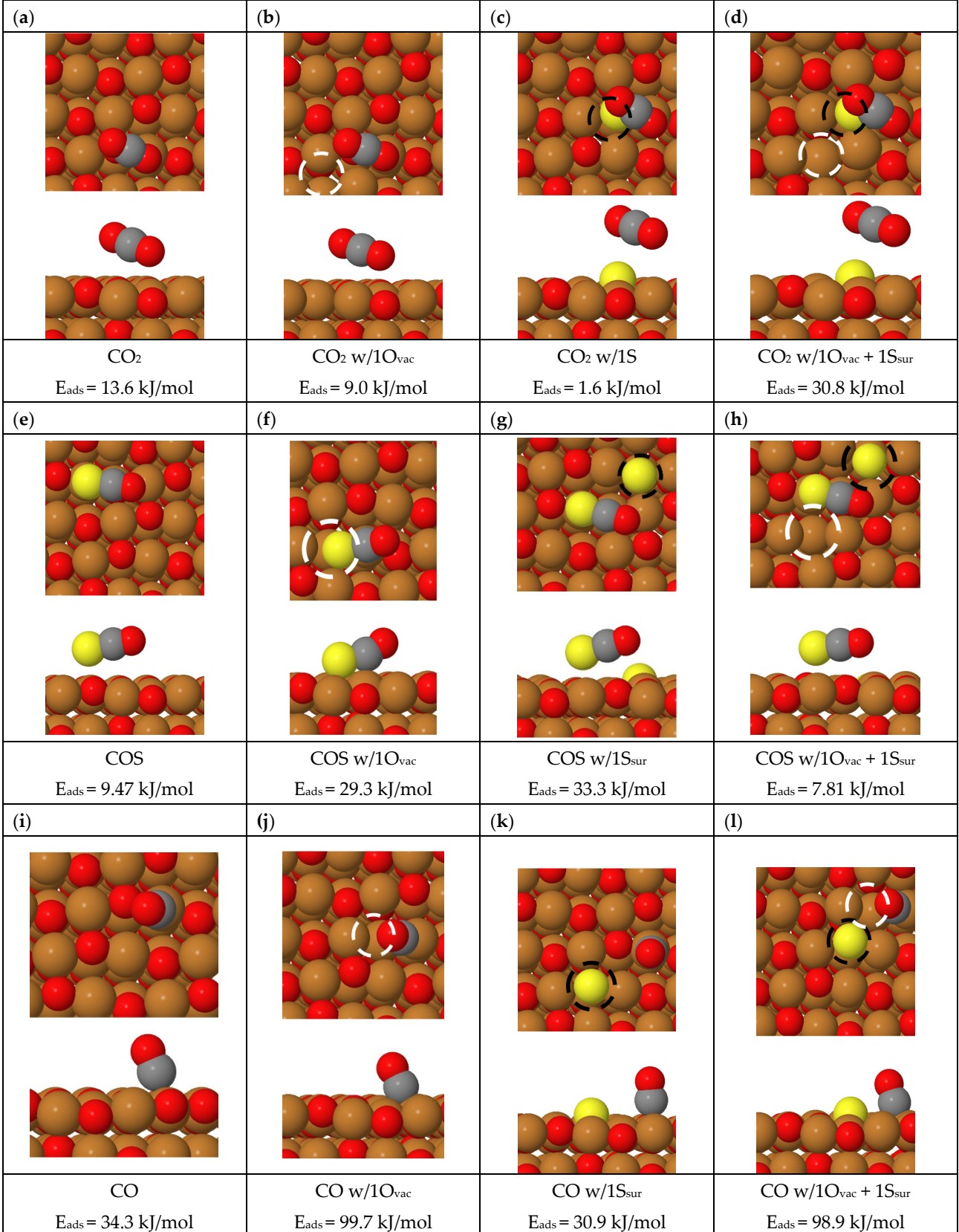

**Figure 11.** Top and side views of adsorbed $CO_2$, COS, and CO on CuO(111) surface with and without an oxygen vacancy and a surface sulfur (yellow). The DFT predicted adsorption energy is presented below each figure.

In terms of the sulfurized surface modeled by replacing surface O atom with S atom, the relaxed adsorbate images are provided in Figure 11c,g,k. A black dashed circle indicates the S surface insulted sites. The surface S atom does not significantly impact the stability of adsorbed $CO_2$* (1.6 kJ/mol) and CO* (30.9 kJ/mol). On the other hand, it stabilizes the COS* (33.3 kJ/mol) on the surface. Finally, adsorption behaviors of the adsorbates on the surface with both a surface oxygen vacancy and a surface sulfur atom were studied. Figure 11d,h,l shows the relaxed adsorbate images. The stability of $CO_2$*(30.8 kJ/mol) was enhanced by ~30 kJ/mol when both surface defect and S surface atom were present, but COS* (7.81 kJ/mol) still binds weakly to the surface. In addition, the stronger adsorption energy of CO* (98.9 kJ/mol) was predicted with the presence of both surface defect and a surface sulfur atom, and the predicted stabilization was similar to the stabilization by an adjacent oxygen vacancy. Overall, the results of $CO_2$* and COS* adsorption do not vary significantly with respect to the presence of surface defect or surface S atom on CuO(111) surface. On the other hand, adsorbed CO* is strongly stabilized by the local environment. The local environment, including the defect sites and the sites adjacent to the surface sulfur, activates the adsorbed molecules (especially CO*), which would strongly affect the kinetics of the CO–S bond dissociation from COS* to CO*.

*5.2. Mechanism of the Initial Desulfurization Kinetics*

Desulfurization kinetics must include two surface reactions, such as CO–S bond-breaking from COS and subsequent $CO_2$ formation. The two reactions proceed sequentially, and the reaction steps are described as follows:

$$COS(g) + * \rightarrow COS^* \qquad (step\ 1)$$
$$COS^* + 1O_{vac}\ site \rightarrow CO^* + S^* \qquad (step\ 2)$$
$$CO^* + S^* + O_{surf} \rightarrow CO_2^* + S^* \qquad (step\ 3)$$

Initially, gas-phase COS adsorbs on an active site *, and the COS* undergoes CO–S bond dissociation into CO* and S*, which occupies a surface oxygen vacancy. The dissociated CO* then grabs a surface oxygen atom to form $CO_2$ on the surface. The described three steps were evaluated on three different CuO(111) surface local environments (1. pristine surface, 2. defected surface with one surface oxygen vacancy, and 3. surface with both surface sulfur and surface oxygen vacancies). All surface environments initially included an additional oxygen vacancy on the surface to provide a site for the dissociated S, which is a $1O_{vac}$ site in the reaction step 2. The pristine CuO surface, for example, has one oxygen vacancy and a different terminology from the pristine CuO surface in the earlier section. Figure 12 provides DFT energy diagrams of COS desulfurization into $CO_2$ and S in the three surface environments. The corresponding relaxed configuration for each step is provided in the Supplemental Information. The DFT calculation predicts that C-S bond dissociation from CO–S requires a high energy barrier of 227.0 kJ/mol with an exothermicity of 55.6 kJ/mol on the pristine CuO(111). Although this initial step is exothermic, the large kinetic barrier would hinder CO–S bond cleavage on the pristine surface. The next step, step 3, was predicted to be endothermic with a high energy barrier of 264.9 kJ/mol to form $CO_2$. Overall, the desulfurization mechanism requires large energies to be activated on the pristine surface, which does not agree with the experimentally observed high performance of CuO at low temperatures. On the defect surface, the COS dissociation step needs to overcome a high-energy barrier of 238.4 kJ/mol with even higher exothermicity (167.7 kJ/mol). DFT predicts that the high reaction barrier (227.0 kJ/mol) for step 3 to form $CO_2$, which is similar to the pristine CuO(111) surface, does provide low endothermicity (34.9 kJ/mol). Zhao et al. also explored the desulfurization mechanisms on CuO(111) surface with surface defects [38]. The reported COS* adsorption energy was 136.23 kJ/mol on an oxygen vacancy introduced on a CuO(111) surface. The large adsorption energy is approximately five times larger than the reported COS* adsorption energy in this study. The large difference stems from the formation of COS-O*, which was defined as COS* in the earlier work by Zhao et al. Such formation requires additional energy to remove the C–O

bond to allow COS* to leave from the COS-O*. Lastly, both sulfur and oxygen vacancies were introduced on the surface to investigate the cooperative effects of surface sulfur and surface oxygen on the desulfurization mechanism. The DFT simulation predicts that step1 would be a barrierless reaction with the reverse barrier of 107.1 kJ/mol when introducing both a sulfur surface atom and an oxygen vacancy. Hence, COS* readily undergoes CO–S bond breaking as soon as COS(g) adsorbs on the surface due to the negligible barrier. In contrast to the effects on CO–S bond breaking, the presence of surface sulfur and oxygen vacancy does not have significant impacts on $CO_2$ formation because DFT predicts still a high reaction barrier (231.4 kJ/mol) with 214.1 kJ/mol endothermicity.

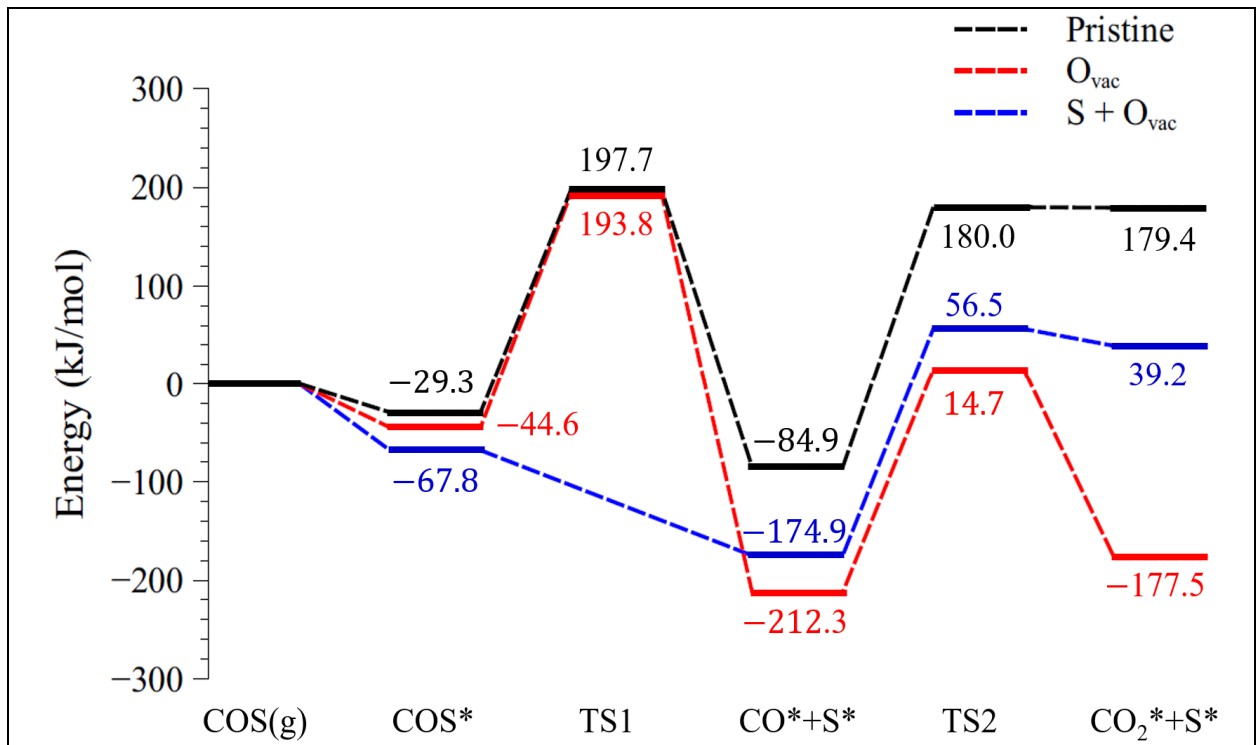

**Figure 12.** DFT evaluated COS oxidation reactions toward $CO_2$ on CuO(111) with and without the oxygen vacancies and a surface sulfur atom. The black, red, and blue lines are the reactions on pristine CuO(111), oxygen vacant employed CuO(111), and both surface sulfur atom and oxygen vacancy introduced CuO(111), respectively. TS stands for transition state.

Although the second step requires a large kinetic barrier in the presence of oxygen vacancy and a surface sulfur atom, such a surface environment potentially accelerates the desulfurization rate. Specifically, the concentration of S* increases as the reaction proceeds on the surface, and it would subsequently combine with the surface defects to form a local environment activating CO–S bond cleavage. As a result, the reaction rate of CO–S bond cleavage increases, resulting in an increase in the concentration of CO*. The increased CO* concentrations would enhance $CO_2$ formation rate, thereby triggering rapid desulfurization on the CuO(111) surface. These simulation results provided initial insights into the roles of the local environment of sulfurized CuO with defects in the desulfurization of COS(g). An in-depth study of the effects, including the effects of subsurface sulfur, will be needed.

### 5.3. Kinetics of CO Production

The CO* adsorption energies were compared with the energy barriers of desulfurization reactions to understand the origin of the experimentally observed CO production during the desulfurization process. The desulfurization reactions are predicted to be less favorable than CO desorption. The CO* adsorption energies ranged from 34 to 100 kJ, which are much lower energy requirements than the energy to activate the desulfurization kinetics (~200 kJ/mol). The high stabilities of CO* on the defect surface and the

surface with both defects and surface sulfur do not have sufficient adsorption energies to remain on the surface compared to the energy barriers for desulfurization. As a result, CO* would experience desorption rather than $CO_2$ formation, which corresponds well to the experimental observation of CO production during the desulfurization reactions. Such behavior would be more pronounced when the surface defect and surface sulfur atom are present because the initial step of desulfurization generating CO* has barrierless exothermic kinetics. Such favorable kinetics would accelerate the generation of CO* on the surface, which leads to facile CO desorption. The CO* production kinetics would occur more frequently at low temperatures because the reaction rates are more clearly distinguished at low temperatures. A future experimental and computational study will conduct an in-depth study of the CO generation kinetics on CuO surfaces to confirm the catalytic activity of CuO for CO production.

## 6. Conclusions

In this study, macro-porous alumina was applied as a support to improve gas diffusion to the absorbent particles. Macro-porous alumina was prepared by moisture removal and heat treatment of a mixed solution containing PMMA colloidal beads and an alumina precursor solution. A sulfur absorbent was prepared by impregnating CuO with excellent sulfur absorption ability into the macro-porous alumina and commercial alumina. The experimental results suggest that the sulfur capacity of the CuO/macro-porous alumina is larger than the CuO/$\gamma$-$Al_2O_3$. The enhancement was attributed to the macro-porous alumina support improving the gas diffusion resistance, allowing the inner unreacted active sites to be activated. Such effects become more significant on the CuO absorbents in the pellet-type supports. Interestingly, CO production was also observed during the desulfurization reaction of COS at a low temperature of ~473 K.

DFT calculations were conducted to investigate the surface reaction of COS desulfurization on the CuO(111) surface. The DFT simulation predicts that the surface kinetics would be affected significantly by the local surface environment of oxygen vacancies and surface sulfur. In particular, the surface local environment for the presence of surface oxygen vacancies and surface sulfur, which is potentially formed during the reaction, makes CO–S bond-breaking significantly facile, which would accelerate the desulfurization rate. Moreover, the CO* stability is lower than the $CO_2$ formation energy barrier. This suggests that CO* would preferably undergo CO desorption rather than $CO_2$ formation, which agrees well with the experimental observation.

Based on the experimental and computational results, a high sulfur capacity was achieved using the macro-porous alumina support, which enhanced the diffusion resistance. Overall, these findings provide a fundamental understanding of the conventional substitution reaction and CO production reaction during the desulfurization reactions on CuO. From the findings of CO production, the Cu-based absorbent can be a catalyst for producing CO(g) from COS(g) dissociation, which would be an interesting topic for CO generation from gasified petroleum coke.

**Supplementary Materials:** The following are available online at https://www.mdpi.com/article/10.3390/app11177775/s1, the images of relaxed configurations for NEB calculations.

**Author Contributions:** Conceptualization, M.K. and N.-K.P.; methodology, N.-K.P., D.B. and Y.J.K.; software, M.K. and D.K.; validation, M.K. and N.-K.P. and D.B.; formal analysis, D.B. and Y.J.K.; investigation, D.K. and D.B.; resources, S.J.L., J.W.L. and Y.Y.; data curation, D.K., D.B. and Y.Y; writing—original draft preparation, D.K. and D.B; writing—review and editing, M.K. and N.-K.P.; supervision, M.K., N.-K.P. and J.W.L.; project administration, S.J.L., J.W.L. and Y.Y.; funding acquisition, M.K. and N.-K.P. All authors have read and agreed to the published version of the manuscript.

**Funding:** This work was supported by a grant (21PCHG-C163217-01) from "Development of Demonstration-scale Hydrogen Production Technology using Petroleum coke" Program funded by Ministry of Land, Infrastructure and Transport of Korea government, and 2020 Yeungnam University Research Grant.

**Institutional Review Board Statement:** Not Applicable.

**Informed Consent Statement:** Not Applicable.

**Data Availability Statement:** Not Applicable.

**Acknowledgments:** We also thank the Ohio Supercomputing Center for providing computational resources.

**Conflicts of Interest:** The authors declare no conflict of interest.

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
