# Peer review of "Enhancement of Desulfurization Capacity with Cu-Based Macro-Porous Sorbents for Hydrogen Production by Gasification of Petroleum Cokes"

_applsci, doi:10.3390/app11177775_

Round 1

Reviewer 1 Report

The work is well-written and the results are meaningful and scientifically relevant. I have the following comments that need to be addressed before I can recommend the acceptance of this study:

  1. References need to be updated. There is only one work included from 2021. This type of work is fairly common and authors can find more relevant papers published in 2021 to cite for the sake of completeness.
  2. There is no dispersion correction included in the DFT calculations. How can the authors justify not correcting for long range effects when they are dealing with the adsorption of a small atom on the surface? The dispersion corrections need to be made here.
  3. There are no zero point energy corrections included for the adsorption energy calculations either. Why not? These effects can alter the results significantly and the authors need to justify why they have skipped this step?

Author Response

We had attached response letter for reviewers comments. We also have done some minor English corrections.  

Reviewer 2 Report

Manuscript ID: applsci-1348223

Title: Enhancement of Desulfurization Capacity with Cu-based Macro-porous Sorbents for Hydrogen Production by Gasification of Petroleum Cokes

Reviewer’s comment

This paper deals with the desulfurization capacity with Cu-based macro-porous sorbents for hydrogen production. Some interesting data is included, and treatment of experimental data and description are proper. Reviewer’s evaluation is that the paper is publishable with minor revisions.

1) line 117 in page 3

Is the mixing ratio of the precursor and the PMMA by weight or in volume ?

2) Figure 3 in page 4

Please add the scale into the images in Fig.3.

3) Figure 6 in page 8

Do you have the crystallite size of CuO on γ-Al2O3 and macro-Al2O3?

4) Table 1

Please explain that the relation between the particle size 200 nm of PMMA and the pore diameter 5.9-6.3 nm of Cu/macro-porous Al2O3.

Reviewer would like to understand the effect of adding the PMMA on the smaller total pore volume and the smaller pore diameter.

Do you have the surface area, pore volume and pore diameter of CuO on γ-Al2O3 and macro-Al2O3 without CuO?

5) lines 290-292 in page 11

Authors consider that the low performance of CuO/ γ-Al2O3 is due to the low gas diffusion resistance. Do you have the pressure drop of the CuO/ γ-Al2O3 and CuO/macro-porous Al2O3?

6) line 358 in page 13

A yellow dashed circle ⇒ A black dashed circle

Author Response

(The authors gave the same response as above.)
